# Partial Soft-Matching Distance for Neural Representational Comparison with Partial Unit Correspondence

**Chaitanya Kapoor**
Department of Cognitive Science
University of California, San Diego
La Jolla, CA, 92093
chkapoor@ucsd.edu

**Alex H. Williams**
Center for Neural Science, New York University
Center for Computational Neuroscience, Flatiron Institute
New York City, NY, 10003
alex.h.williams@nyu.edu

**Meenakshi Khosla**
Department of Cognitive Science
Department of Computer Science and Engineering
University of California, San Diego
La Jolla, CA, 92093
mkhosla@ucsd.edu

## Abstract

Representational similarity metrics typically force all units to be matched, making them susceptible to noise and outliers common in neural representations. We extend the soft-matching distance to a partial optimal transport setting that allows some neurons to remain unmatched, yielding rotation-sensitive but robust correspondences. This partial soft-matching distance provides theoretical advantages—relaxing strict mass conservation while maintaining interpretable transport costs—and practical benefits through efficient neuron ranking in terms of cross-network alignment without costly iterative recomputation. In simulations, it preserves correct matches under outliers and reliably selects the correct model in noise-corrupted identification tasks. On fMRI data, it automatically excludes low-reliability voxels and produces voxel rankings by alignment quality that closely match computationally expensive brute-force approaches. It achieves higher alignment precision across homologous brain areas than standard soft-matching, which is forced to match all units regardless of quality. In deep networks, highly matched units exhibit similar maximally exciting images, while unmatched units show divergent patterns. This ability to partition by match quality enables focused analyses, *e.g.,* testing whether networks have privileged axes even within their most aligned subpopulations. Overall, partial soft-matching provides a principled and practical method for representational comparison under partial correspondence.

## 1 Introduction

Understanding how design choices (*e.g.,* training objectives, architecture) shape neural representations requires comparing how different systems encode information. A fundamental challenge in this comparison is determining which computational units correspond across systems: do specific neurons implement similar functions across networks? This is central to understanding whether different systems converge to similar computational solutions. Most existing representational similarity metrics, such as CKA (Kornblith et al., 2019), RSA (Kriegeskorte et al., 2008), Procrustes distance (Williams et al., 2021; Ding et al., 2021), and CCA variants (Raghu et al., 2017), are rotation-invariant—they measure geometric similarity while ignoring the specific axes along which

---

All code is publicly available at: https://github.com/NeuroML-Lab/partial-metric/

information is encoded. This limitation prevents us from understanding neuron-level correspondence and whether systems share axis-aligned representations.

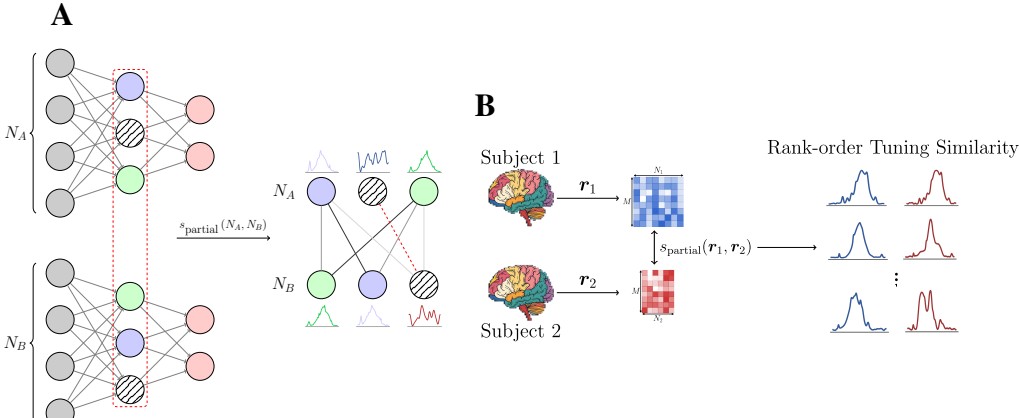

Figure 1: **Partial Soft-Matching Distance for Matching Tuning Curves.** **(A)** Two toy networks $N_A$ and $N_B$; the layer of interest for alignment is shown in red. A partial matching recovers one-to-one correspondences between units with highly similar tuning curves. Line color encodes match strength (darker = stronger). By contrast, a purely soft-matching yields a spurious pair (hatched units, red dotted line). **(B)** The same metric can be used to rank voxel/unit tuning-curve similarity between two subjects' responses $\{r_1, r_2\}$, when exposed to the same visual stimulus.

To address these challenges, Khosla & Williams (2024) recently proposed a metric based on discrete optimal transport (OT; Peyré et al., 2019) called the *soft-matching distance* which finds rotation-sensitive correspondences between neurons while remaining invariant to their ordering. However, this approach requires that all neural units are matched across networks. This requirement may be a crucial limitation in certain settings, since neural populations often contain noisy, inactive, or task-irrelevant units—particularly in biological recordings from fMRI or electrophysiology. Moreover, even task-relevant units may be model-specific, implementing computations unique to a particular architecture or training regime in deep neural networks (DNNs). When comparing networks, we should not expect complete overlap in their functional units. Forcing all units into correspondence may therefore inflate distances and produce misleading alignments.

Here, we introduce the **partial soft-matching distance** (Fig. 1), which allows a fraction of neurons to remain unmatched while preserving robust correspondences among the remainder. Our method provides several key advantages:

- **Theoretical robustness:** Relaxing mass conservation allows the metric to handle populations with different numbers of units, where some may lack correspondence (*e.g.*, due to noise).
- **Computational efficiency:** Achieves comparable rankings with a single $\mathcal{O}(n^3 \log n)$ computation, unlike brute-force methods requiring $\mathcal{O}(n^4 \log n)$ operations.
- **Interpretable partitioning:** Separates well-matched from unmatched units, enabling focused analysis of aligned subpopulations.

We demonstrate these advantages through controlled simulations showing correspondence despite spurious neurons, and accurate model identification in noise-corrupted scenarios. In fMRI data from the Natural Scenes Dataset (Allen et al., 2022), our method discards low-quality voxels and outperforms standard soft-matching in aligning homologous brain regions across subjects. When applied to DNNs, we find that highly-matched units produce similar maximally exciting images (MEIs) across models, while unmatched units show divergent MEIs, suggesting distinct computational roles. Crucially, filtering unmatched units using partial soft-matching improves alignment over heuristics based on soft-matching correlations, matching the performance of a computationally intensive brute-force method that iteratively removes units in a greedy fashion. This framework provides a principled approach for comparing neural representations under partial correspondence—a common scenario in neuroscience and AI.

## 2    METHODS

The optimal transport (OT) problem finds the minimum-cost mapping between probability distributions, yielding metrics like the soft-matching distance (Khosla & Williams, 2024). However, classical OT requires equal total mass between distributions—a constraint violated in neural recordings where units may be noisy, inactive, or genuinely non-corresponding. We extend the soft-matching distance to handle these realistic scenarios through partial optimal transport.

### 2.1    SOFT-MATCHING DISTANCE

Consider two neural populations with $N_x$ and $N_y$ units respectively, each with "tuning curves", $\{\boldsymbol{x}_i\}_{i=1}^{N_x}$ and $\{\boldsymbol{y}_j\}_{j=1}^{N_y}$ taking values in $\mathbb{R}^M$. Each unit's tuning curve, respectively denoted $\mathbf{x}_i$ and $\mathbf{y}_j$ for the two neural populations, represents a neuron's response over a set of $M$ probe stimuli. Stacking these tuning curve vectors column-wise produces matrices $\boldsymbol{X} \in \mathbb{R}^{M \times N_x}$ and $\boldsymbol{Y} \in \mathbb{R}^{M \times N_y}$.

The soft-matching distance treats each population as a uniform empirical measure and quantifies the optimal transport cost between them:

$$d_T(\boldsymbol{X}, \boldsymbol{Y}) = \min_{\boldsymbol{T} \in \mathcal{T}(N_x, N_y)} \sqrt{\langle \boldsymbol{C}, \boldsymbol{T} \rangle_F}$$

where $\boldsymbol{C}_{ij} = \|\boldsymbol{x}_i - \boldsymbol{y}_j\|^2$ is the squared Euclidean transport cost, $\langle \cdot, \cdot \rangle_F$ the Frobenius inner product, and $\mathcal{T}(N_x, N_y)$ is the transportation polytope (De Loera & Kim, 2013), i.e., the set of all $N_x \times N_y$ nonnegative matrices whose rows each sum to $1/N_x$ and whose columns each sum to $1/N_y$.

This formulation is permutation-invariant yet rotation-sensitive, revealing single-neuron tuning alignment. Furthermore, $d_T$ is symmetric and satisfies the triangular inequality, which has been argued to be important for certain analyses of neural representations (Williams et al., 2021; Lange et al., 2023). The key limitation (which we document in Sections 3 and 4) is that the marginal constraints (i.e. that the transport plan $\boldsymbol{T}$ lie within the transportation polytope) forces all units to be matched, producing spurious correspondences when populations contain non-corresponding units.

### 2.2    PARTIAL SOFT-MATCHING DISTANCE

The soft-matching formulation requires the two empirical distributions to have identical total mass and further enforces that **all** mass must be transported. The partial OT problem extends this by allowing only a pre-specified fraction $0 \leq s \leq 1$ of the total mass to be matched at minimal cost.

Formally, for empirical measures with unit total mass, a natural set of admissible couplings is

$$\mathcal{T}^s(N_x, N_y) = \left\{ \boldsymbol{T} \in \mathbb{R}_+^{N_x \times N_y} \,\middle|\, \sum_{j=1}^{N_y} \boldsymbol{T}_{ij} \leq \tfrac{1}{N_x}, \quad \sum_{i=1}^{N_x} \boldsymbol{T}_{ij} \leq \tfrac{1}{N_y}, \quad \sum_{i,j} \boldsymbol{T}_{ij} = s \right\}.$$

Here, the inequalities on the row/column marginals permit mass to remain unmatched in either population, and the scalar $s$ controls the total matched mass. Since we normalize our populations to have unit total mass, $s$ directly represents the fraction of units that are actually matched. The partial soft-matching distance is then the minimum transport cost over this feasible set,

$$d_{\boldsymbol{T}}(\boldsymbol{X}, \boldsymbol{Y}\boldsymbol{X}) = \min_{\boldsymbol{T} \in \mathcal{T}^s(N_x, N_y)} \langle \boldsymbol{C}, \boldsymbol{T} \rangle_F,$$

with $\boldsymbol{C}$ the usual cost matrix (e.g., $\boldsymbol{C}_{ij} = \|\boldsymbol{x}_i - \boldsymbol{y}_j\|^2$). In our formulation, we use pairwise cosine distance as the cost function. Several numerical approaches have been developed to solve partial OT problems (Benamou et al., 2015; Chizat et al., 2018). More recently, Chapel et al. (2020) augmented the cost matrix with dummy (or virtual) points which are assigned large transportation cost. All mass routed to these dummy nodes is effectively discarded, which yields an exact partial-matching solution in the augmented formulation.

Partial OT distances do not satisfy the triangle inequality and therefore are not proper metrics. However, they still provide a symmetric notion of dissimilarity in representation and, as we document in Sections 3 and 4, they provide robust and interpretable tool for tuning-level comparisons between neural populations with unequal or noisy measurements.

## 2.3 Choosing Optimal Regularization

To apply partial soft-matching in practice, we must select the hyperparameter $0 \leq s \leq 1$ which determines how much mass to transport between distributions. This is a key challenge when the abundance of outliers and magnitude of noise in the data are unknown *a priori*. To address this, we adopt an L-curve heuristic (Cultrera & Callegaro, 2020), inspired by classical regularization methods for ill-posed problems (*e.g.,* Tikhonov regularization). The L-curve captures the trade-off between transport distance and regularization strength, with the "elbow" typically indicating a balanced choice between these competing objectives. Concretely, we define the two-dimensional parametric curve:

$$f(s) = (\zeta(s), \rho(s)) \rightarrow \begin{cases} \zeta(s) = \langle \boldsymbol{T}(s), \, \boldsymbol{C} \rangle_F \\ \rho(s) = 1 - s \end{cases}$$

where $\boldsymbol{C}$ is the cost matrix and $\boldsymbol{T}(s)$ is the optimal transport plan for a match fraction $s \in [0, 1]$. We interpret $\rho(s)$ as the regularization strength—smaller $s$, or conversely larger $\rho$ permits more mass to be left unmatched. The optimal regularization $s_0$ is identified at the curve's point of maximal positive curvature (the elbow), which balances low transport cost against aggressive regularization.

In our discrete implementation, we sample $s$ uniformly from a sequence $\{s_i\}_{i=1}^N$ and compute the associated transportation costs $\zeta_i = \zeta(s_i)$. We compute the elbow by approximating the second derivative of the cost curve with respect to the regularization strength $\rho(s)$ by the centered second finite difference $\delta_\rho^2$,

$$\delta_\rho^2 \zeta_i = \zeta_{i+1} - 2\zeta_i + \zeta_{i-1} \ \text{ for } \ i = 2, \ldots, N-1$$

and select the index with maximal positive curvature

$$i^\star = \underset{2 \leq i \leq N-1}{\arg\max} |\delta^2 \zeta_i|, \quad s_0 = s_{i^\star}$$

allowing us to analytically select the optimal regularization $s_0$.

## 2.4 Partial Soft-Matching as a Correlation Score

Suppose that the tuning curves in two neuron populations $\boldsymbol{X}$ and $\boldsymbol{Y}$ have been mean-centered and scaled to unit-norm. Under this normalization, the inner product $\boldsymbol{x}_i^\top \boldsymbol{y}_j$ is identical to the Pearson correlation between neuron $i$ in $\boldsymbol{X}$ and neuron $j$ in $\boldsymbol{Y}$. Using this, the optimization can now be recast as a *maximization* of total matched correlation:

$$d^{\text{corr}}(\boldsymbol{X}, \boldsymbol{Y}) = \max_{\boldsymbol{T} \in \mathcal{T}^s(N_x, N_y)} \sum_{ij} \boldsymbol{T}_{ij} \boldsymbol{x}_i^\top \boldsymbol{y}_j$$

Intuitively, $d^{\text{corr}}$ measures the average correlation between paired neurons under the coupling $\boldsymbol{T}$. We report $d^{\text{corr}}$ for the remainder of the manuscript. We also report alignment obtained using a squared Euclidean cost function, $\boldsymbol{C}_{ij} = ||\boldsymbol{x}_i - \boldsymbol{x}_j||^2$, in Appendix A1.8 and observe identical results.

## 2.5 Interpretation and Output

The optimal transport plan $\boldsymbol{T}^\star$ provides a soft partial alignment where:

- Row sums $\in [0, 1/N_x]$: amount of mass transported from each source neuron

- Column sums $\in [0, 1/N_y]$: amount of mass received by each target neuron

- Total transported mass equals $s < 1$ (the fraction of total mass matched)

- Near-zero row/column sums identify effectively unmatched units

This partitions populations based on participation in the optimal matching, from completely unmatched to maximally participating units.

# 3 SIMULATIONS: ROBUSTNESS TO NOISE AND SELECTING THE "CORRECT" MODEL

We designed controlled simulations to evaluate whether partial soft-matching **(1)** maintains accurate correspondences despite spurious neurons and **(2)** correctly identifies which model shares more signal with a reference population. Synthetic neural representation generation is detailed in Appendix A1.3.

## 3.1 ROBUSTNESS AGAINST SPURIOUS NEURONS

We construct two neural populations $X$ and $Y$, each containing $K$ "signal" neurons matched pairwise. We introduce noise by augmenting $X$ with $M_x$ random neurons and $Y$ with $M_y$ random neurons, where each random neuron is drawn from $\varepsilon \sim \mathcal{N}(0, 1)$. The resultant populations are thus $X \in \mathbb{R}^{(K+M_x) \times N}$ and $Y \in \mathbb{R}^{(K+M_y) \times N}$, where $N$ is the number of unique stimuli.

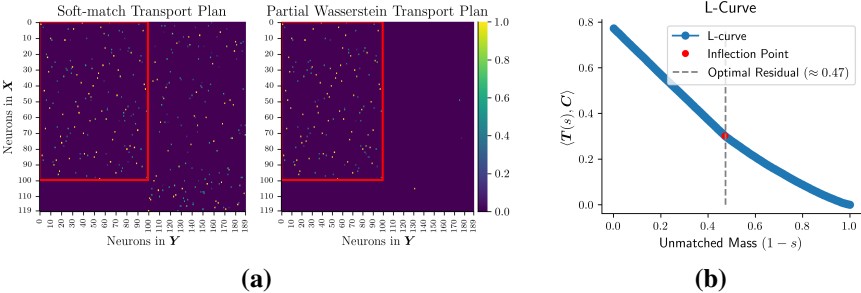

**(a)**                                                                                        **(b)**

Figure 2: **Comparison of Balanced and Partial Soft-Matching.** **(a)** We simulate two neural representations, $X$ and $Y$, with 120 and 190 neurons respectively. The first 100 neurons represent pure *signal*, while the rest are pure *noise*. Red denotes all pure signal neurons in the two representations. **(b)** The L-curve method selects the optimal mass regularization parameter ($= 90/190 \approx 0.47$), successfully discarding noisy units.

In the (fully balanced) soft-matching distance must match *all* $K + M_x$ neurons to $K + M_y$ neurons. This forces spurious outlier-to-outlier assignments, which inflate the overall transport cost and, consequently, the distance. In contrast, the partial soft-matching distance only transports mass corresponding to $K$ true matches, ignoring the random neurons. As a result, the recovered transport cost is significantly smaller and reflects the true correspondence between the signal neurons. We visualize the transport plans for both—soft-matching and partial soft-matching in Fig. 2, and observe that the L-curve heuristic is able to faithfully distinguish between noise and signal units.

## 3.2 CHOOSING BETWEEN TWO MODELS

Suppose we consider two models: *Model A*, where $Y_a$ shares exactly $K$ correctly matched neurons with $X$, along with $M_y$ additional noisy neurons; and *Model B*, where $Y_b$ **(i)** does not contain the same signal neurons as $X$, and **(ii)** shares fewer correctly matched neurons.

*Model A* is considered "correct" here because it preserves the maximum number of genuine signal correspondences with $X$—the $K$ matched neurons encode the same computational features as their counterparts in $X$—plus additional noisy neurons. These extra neurons may reflect measurement noise, inactive recording channels, or recording artifacts that are common in real neural data. *Model B*, in contrast, shares only a subset of $X's$ signal neurons.

We compute the partial soft-matching scores $s_{\text{partial}}(X, Y_a)$ and $s_{\text{partial}}(X, Y_b)$. Because partial OT can ignore outliers and preserve only the true $K$ matches, the distance, to the *"correct"* model $Y_a$ will be significantly smaller—equivalently, the correlation satisfies $s_{\text{partial}}(X, Y_a) > s_{\text{partial}}(X, Y_b)$, correctly identifying *Model A* as sharing more signal with $X$. By contrast, standard soft-matching forces matches for all units (including noise), obscuring signal differences and failing to discriminate $Y_a$ from $Y_b$, as shown in Fig. 3.

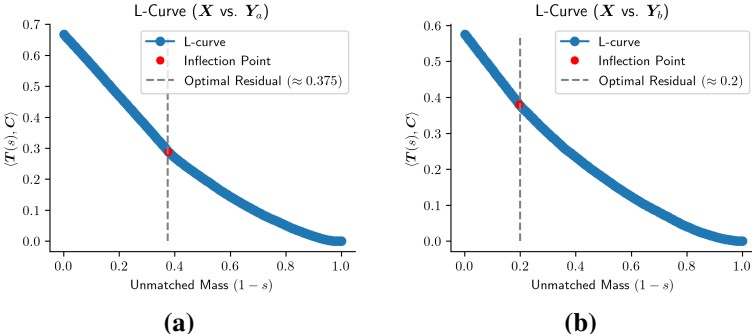

Figure 3: **Model Selection Using Partial Soft-Matching.** We simulate three synthetic representations to test whether partial soft-matching correctly identifies which of two candidate models—$Y_a$ or $Y_b$—shares more signal with a reference population $X$ (100 units). $Y_a$ contains all 100 signal units from $X$ plus 60 noise units; $Y_b$ contains 100 units, 80 of which match $X$. The true fraction of shared units is known *a priori*, marked by a vertical gray line. With the L-curve-selected regularization, partial soft-matching yields correlation scores $s_{\mathrm{partial}}(X, Y_a) = 0.715$ and $s_{\mathrm{partial}}(X, Y_b) = 0.645$, correctly favoring $Y_a$. Standard soft-matching fails, with $s_{\mathrm{sm}}(X, Y_a) = 0.339$ and $s_{\mathrm{sm}}(X, Y_b) = 0.415$, incorrectly preferring $Y_b$ due to forced matching of noise.

## 4 APPLICATIONS IN NEUROSCIENCE AND AI

### 4.1 COMPARISONS OF NEURAL RECORDINGS ACROSS SUBJECTS

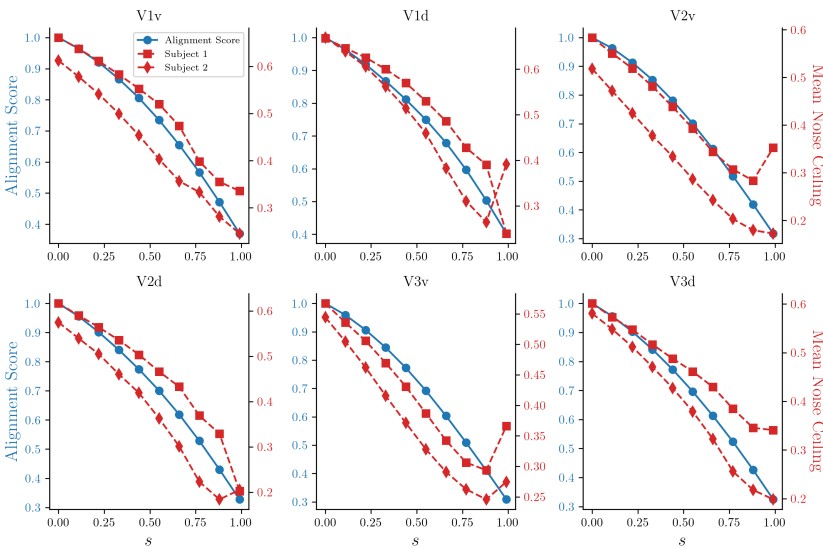

Figure 4: **Aligning Voxel Responses Between Different Subjects in NSD.** For each area, we plot the **(i)** partial soft-matching score at different mass regularization values and **(ii)** the mean noise ceilings of the voxels that were kept at that regularization. The alignment criterion consistently identifies low noise-ceiling voxels for exclusion.

Perfect correspondence of neural populations across subjects is rare—measurement noise, inactive voxels and anatomical variability implies imprecise region boundary definition. Voxels nominally assigned to the same brain area can sample neighboring regions implementing distinct computations. Individual differences in functional organization can further aggravate distinct computations across subjects. Partial soft-matching addresses these challenges by selectively excluding non-corresponding units from the alignment.

We demonstrate this on voxel responses from a subject pair (IDs 1 and 2) across six visual areas (V1v, V1d, V2v, V2d, V3v, V3d) from the Natural Scenes Dataset (Allen et al., 2022). Fig. 4 shows how voxel selection quality changes as we vary the mass regularization parameter $s$ from 1 (including all voxels) to 0 (excluding all voxels). As $s$ decreases and we exclude more voxels, the mean noise ceiling of the retained voxels steadily increases, while the alignment score between these retained voxels also improves. Since noise ceiling measures the reliability of a voxel's responses across repeated stimulus presentations, this demonstrates that our method successfully identifies and excludes voxels with poor response replicability. By progressively discarding these unreliable measurements, partial soft-matching automatically focuses the alignment on the subset of voxels that provide the most consistent and well-matched signal across subjects. We perform an identical experiment on a different NSD subject pair (Appendix A1.2) and observe identical results.

## 4.2 Comparison Against Baseline Methods

In this section, we demonstrate the utility of our metric as an efficient tool for rank-ordering neurons by their degree of cross-population alignment. We compare three approaches—brute-force matching, correlation-based ordering and our proposed partial soft-matching method. We test these methods in two distinct settings: **(1)** comparing convolutional kernels between two ResNet-18 models trained from different random initializations on ImageNet (Deng et al., 2009), examining early, middle, and late layers (Fig. 5), and **(2)** aligning voxel responses between human subjects viewing natural images, across six visual areas (V1v, V1d, V2v, V2d, V3v, V3d) from the Natural Scenes Dataset (Fig. 6).

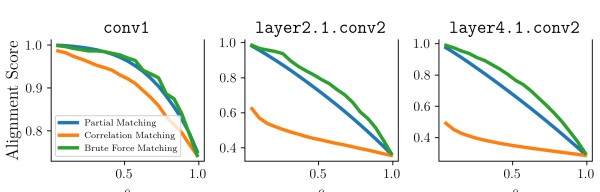

Figure 5: **Evaluating Methods for Identifying (Un)matched Neurons in Deep Networks.** We compare three methods for ranking convolutional kernels by alignment between two ResNet-18 models trained from different random initializations on ImageNet, across early, middle, and late layers. Removing low-alignment units identified by partial soft-matching yields alignment scores nearly identical to those obtained by removing kernels ranked least important via brute-force ablations, while correlation-based rankings perform poorly.

*Brute-force matching* provides the ground-truth ranking by exhaustively testing each neuron's contribution to alignment. We fit an optimal soft-matching transformation to the complete representation, then iteratively remove each neuron and recompute the entire soft-matching optimization to measure the impact on alignment score. This produces an exact ranking of neurons by their alignment quality. However, each soft-matching optimization requires $\mathcal{O}(n^3 \log n)$ operations and for $n$ neurons to test, the total complexity is $\mathcal{O}(n^4 \log n)$, making this approach computationally prohibitive for realistic population sizes.

*Correlation-based ordering* attempts a computationally cheaper approximation by computing pairwise Pearson correlations between neurons using the transport plan from a single soft-matching optimization. As shown in Figures 5 and 6, this heuristic fails catastrophically—it incorrectly identifies and removes neurons that are actually crucial for alignment, resulting in dramatically degraded alignment scores. This failure occurs because individual correlation values don't capture the global optimization structure of the transport problem.

*Partial soft-matching* offers a nuanced tradeoff. To obtain a complete ranking of all $n$ neurons (matching the output of brute-force), we would still require $n$ separate optimizations at different regularization values, maintaining $\mathcal{O}(n^4 \log n)$ complexity. However, for the practically relevant task of identifying the top $X\%$ most-aligned or least-aligned neurons—which suffices for most analyses in neuroscience and deep learning which require identifying highly-aligned or poorly-aligned subpopulations rather than complete rankings—a single optimization at the appropriate regularization value $\left(\mathcal{O}(n^3 \log n)\right)$ provides near-identical results to brute-force ranking. As Figures 5 and 6 demonstrate, when selecting subsets of neurons at various alignment thresholds, our method's

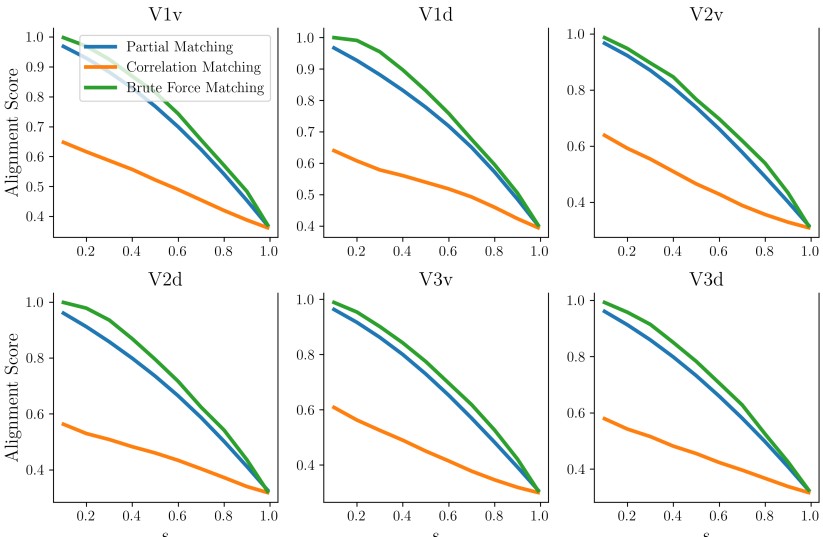

Figure 6: **Evaluating Methods for Identifying (Un)matched Voxels in Brain Data.** We evaluate three methods for ranking voxels by their degree of alignment between a subject pair from NSD across six visual areas. Removing low-alignment voxels identified by partial soft-matching yields alignment scores nearly identical to those obtained by removing voxels ranked least important via brute-force ablations, while correlation-based rankings perform poorly.

selections yield alignment scores nearly matching those from exhaustive brute-force ranking, while correlation-based selection performs poorly. Full algorithmic details are provided in Appendix A1.4.

### 4.3 MAPPING (DIS)SIMILAR BRAIN REGIONS

A robust similarity metric for neural populations should exhibit specificity: it must identify when responses come from the same brain area across subjects (true positives) while avoiding false matches between distinct areas (false positives) (Thobani et al., 2025). This specificity is crucial when anatomical boundaries are imprecise and individual variability is high.

We evaluate this specificity by testing whether partial soft-matching correctly aligns homologous visual areas while maintaining separation between distinct areas. Concretely, we select two visual ROIs in a subject pair and then compute a between-subject matching for all voxels of these regions. For each pair of visual regions within and across subjects from the NSD, we compute the precision of voxel assignments—the fraction of matched voxels that truly belong to corresponding regions. Table 1 shows these precision scores comparing standard soft-matching (which must match all voxels), thresholding using voxel noise ceilings, and partial soft-matching (which can exclude poor correspondences). The optimal regularization parameter for partial soft-matching is chosen via the L-curve heuristic as described in Section 2.3.

Across most region pairs, partial soft-matching achieves higher precision than standard soft-matching and thresholding, with striking improvements for several cross-area comparisons (*e.g.,* V1d + V2v: $0.906 \rightarrow 0.971$). This improvement stems from the method's ability to exclude voxels that lack clear correspondence—whether due to boundary uncertainty or measurement noise. By not forcing these ambiguous voxels into the matching, partial soft-matching maintains cleaner separation between distinct regions while preserving strong alignment within homologous areas.

### 4.4 MAXIMALLY EXCITING IMAGES

Maximally Exciting Images (MEIs)—synthetic stimuli optimized to maximize individual unit responses—provide an interpretable visualization of what each neuron "looks for" in its input (Erhan et al., 2009; Pierzchlewicz et al., 2023; Walker et al., 2019; Bashivan et al., 2019). We synthesize

| Brain Region Pair | SM Precision ($\uparrow$) | ParSM Precision ($\uparrow$) | $\epsilon = 0.1$ ($\uparrow$) | $\epsilon = 0.3$ ($\uparrow$) |
|---|---|---|---|---|
| V1v + V1d | 0.839 | **0.905** (0.76) | 0.847 (0.98) | 0.855 (0.94) |
| V1v + V2v | 0.677 | 0.680 (0.99) | 0.680 (0.96) | **0.695** (0.88) |
| V1v + V2d | 0.880 | 0.884 (0.99) | 0.884 (0.97) | **0.894** (0.91) |
| V1v + V3v | 0.798 | **0.853** (0.97) | 0.803 (0.99) | 0.815 (0.90) |
| V1v + V3d | 0.882 | 0.890 (0.98) | 0.890 (0.97) | **0.913** (0.91) |
| V1d + V2v | 0.881 | **0.971** (0.71) | 0.889 (0.96) | 0.906 (0.89) |
| V1d + V2d | 0.706 | 0.708 (0.99) | 0.720 (0.97) | **0.727** (0.92) |
| V1d + V3v | 0.879 | 0.881 (0.99) | 0.885 (0.97) | **0.892** (0.91) |
| V1d + V3d | 0.803 | **0.878** (0.76) | 0.818 (0.97) | 0.828 (0.92) |
| V2v + V2d | 0.869 | 0.879 (0.98) | 0.880 (0.95) | **0.896** (0.87) |
| V2v + V3v | 0.651 | **0.661** (0.95) | 0.653 (0.95) | 0.654 (0.86) |
| V2v + V3d | 0.853 | 0.856 (0.99) | 0.867 (0.95) | **0.882** (0.87) |
| V2d + V3v | 0.833 | **0.971** (0.42) | 0.845 (0.96) | 0.863 (0.88) |
| V2d + V3d | 0.638 | **0.643** (0.99) | 0.642 (0.96) | 0.643 (0.89) |
| V3v + V3d | 0.814 | 0.822 (0.99) | 0.828 (0.96) | **0.852** (0.88) |

Table 1: **Precision of Cross-Subject Voxel Alignment Within and Across Visual Areas**. Comparison of soft-matching (SM), partial soft-matching (ParSM) and noise ceiling thresholding to align voxels between visual regions in an NSD subject pair. Precision measures the fraction of matched voxels belonging to corresponding anatomical regions (higher = better specificity). We include the fraction of total voxels that contribute towards computing alignment in parenthesis. The $\epsilon$ values denote the noise ceiling threshold below which voxels are excluded. ParSM almost always yields higher precision by excluding voxels that lack clear correspondence.

MEIs[1] for unit pairs from two ResNet-18 models trained with different random seeds, sampling from neurons ranked as highly-matched (top $10\%$ of transport mass) versus poorly-matched (bottom $10\%$) by our metric. Fig. 7 shows striking differences: highly-matched pairs produce nearly identical MEIs, revealing that these units have converged on similar feature detectors despite independent training. In contrast, unmatched pairs yield divergent MEIs with distinct visual patterns, confirming they likely implement different computations. We demonstrate this with some additional results in Appendix A1.7.

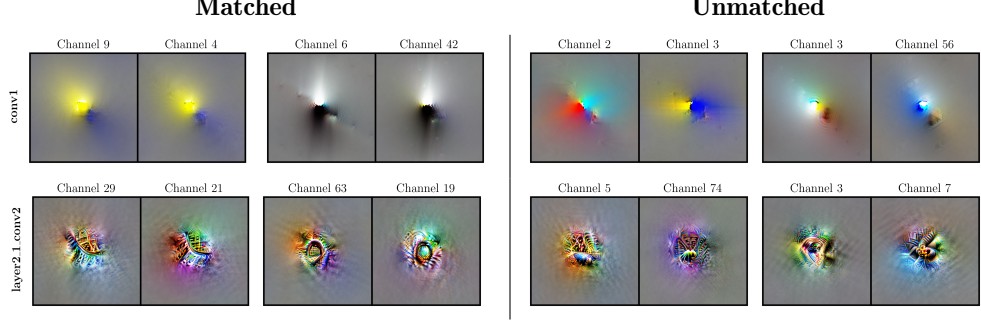

Figure 7: **Visualization of (Un)matched Units Using Maximally Exciting Images.** We show MEIs for two layers of ResNet-18, comparing units classified as matched or unmatched by the partial soft-matching metric. Matched examples are sampled from the top $10\%$ of partial soft-matching scores; unmatched examples are sampled from the bottom $10\%$.

## 4.5 Testing for privileged axes in aligned neural subpopulations

Neural networks could in principle encode information in arbitrarily rotated coordinate systems, yet recent evidence suggests they converge on specific "privileged" axes. Networks trained from different initializations not only share representational geometry but actually align their coordinate

---

[1]Appendix A1.5

systems—with individual neurons implementing similar computations across networks (Khosla & Williams, 2024; Khosla et al., 2024; Kapoor et al., 2025). This privileged basis hypothesis suggests that certain directions in activation space are preferred, potentially due to architectural constraints, *e.g.,* axis-aligned nonlinearities (ReLU). We ask whether this alignment holds when restricted to the best-matched neurons. Using partial soft-matching, we partition increasingly well-matched neurons and test for privileged axes from the full population down to the strongest matched pairs.

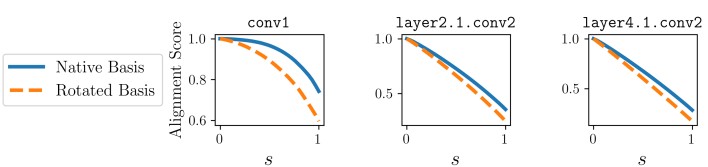

For two ImageNet-trained ResNet-18 models initialized with different random seeds, we extract representations $\{\boldsymbol{X}_1, \boldsymbol{X}_2\}$ at each convolutional layer. To test for privileged axes, we apply a random orthogonal transformation $\boldsymbol{Q}$ (sampled uniformly from the Haar measure) to one network's

Figure 8: Alignment between ResNet-18 models under original and randomly rotated coordinate systems, across early, middle and later layers and matched-mass thresholds. Rotation reduces alignment at all thresholds—including among the best-matched units—supporting convergence to a shared coordinate system even among the most aligned subpopulations.

representation and measure how this rotation affects alignment $s_{\mathrm{partial}}(\boldsymbol{X}_1\boldsymbol{Q}, \boldsymbol{X}_2)$. If neurons were arbitrarily oriented, this rotation would not affect the alignment. However, if a privileged basis exists, rotating away from it should decrease alignment.

We test multiple regularization values ($s$) to sample subpopulations of varying alignment quality. Fig. 8 reveals that alignment consistently decreases under rotation across for all $s$ and depths, with an identical pattern across all convolutional layers (Appendix A1.6). This demonstrates that privileged coordinate systems persist even among the most aligned neural subpopulations—the subset we might expect to be most robustly matched. The persistence of this effect suggests that coordinate alignment is not merely a statistical artifact of analyzing many neurons together, but reflects true convergence on similar computational solutions at the single-unit level.

## 5 DISCUSSION

We introduced partial soft-matching, extending OT-based representational comparisons to account for partial correspondence between neural populations. This addresses a key limitation of methods that force all units into alignment, which can obscure genuine matches in the presence of noise. Simulations show that the method preserves true correspondences under noise and selects the correct model in system identification tasks. In fMRI data, it excludes low-reliability voxels and improves alignment precision across homologous brain regions. In deep networks, matched units exhibit shared MEIs, while unmatched units differ qualitatively. In both domains, partial soft-matching provides a more efficient way to order units by alignment quality, closely matching brute-force ablations, requiring only a single optimization at each chosen mass regularization value.

Some limitations remain. The L-curve heuristic for selecting matched mass performs well empirically, but its generality is unclear. We list some good practices that a practitioner should keep in mind while using the L-curve method in Appendix A1.1. Alternate strategies (*e.g.,* area under the alignment-regularization curve)—may offer more robust summarization across multiple regularization values. Because partial OT relaxes mass conservation and violates the triangle inequality, it should be understood as a comparative tool rather than a strict metric. However, we note that recent theoretical work has developed partial Wasserstein variants that preserve full metric properties, including the triangle inequality (Raghvendra et al., 2024). Future extensions could integrate these formulations for applications requiring strict metric axioms, such as clustering analyses. Although significantly faster than brute-force baselines, the $\mathcal{O}(n^3 \log n)$ cost can limit scalability to very large datasets. These considerations aside, this work highlights that meaningful comparison does not require complete unit overlap: partial soft-matching enables principled analysis of convergent and divergent representational structure across neural systems.

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
