# A1 APPENDIX

## A1.1 BEST PRACTICES

We treat the L-curve elbow as a practical heuristic for selecting the matched mass $s$, rather than a formal rule. As with any hyperparameter in machine learning, the L-curve should be treated as a user-dependent choice.

The L-curve typically fails when the cost-regularization curve $(\zeta(s), \rho(s))$ is smooth and monotonic. In this regime, the estimated curvature is uniformly small, and any computed *"inflection"* is likely an artifact of either numerical noise or local smoothness. A common empirical signature of this failure mode is that the algorithm selects an "optimal" regularization $s$ at either of the tail ends of the $(\zeta, \rho)$ curve. When this occurs, we suggest a simple diagnostic—one can visually inspect the L-curve and check the magnitude of curvature at the selected point. However, if the curvature profile is non-informative, one can use alternative rank summary statistics such as area under the $(\zeta, \rho)$ curve.

## A1.2 COMPARISON OF NEURAL RECORDINGS FOR AN ALTERNATE SUBJECT PAIR

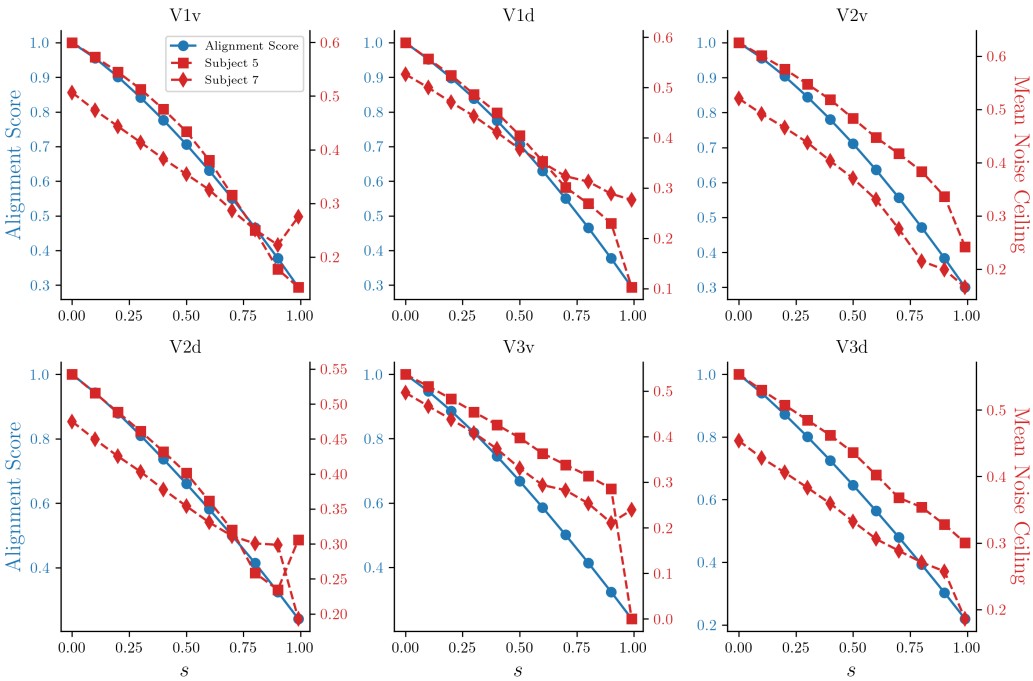

Figure A1: **Aligning Voxel Responses Between Subject IDs 5 and 7 in NSD.** We plot the partial soft-matching alignment score achieved at different mass regularization values and the mean noise ceilings of the voxels that were kept at each regularization. For The alignment criterion consistently identifies low noise-ceiling voxels for exclusion for Subjects 5 and 7.

## A1.3 GENERATION OF SYNTHETIC DATA

For the synthetic experiments in Section 3, we construct two approximately one-to-one matched "neural representations" by linearly mixing a small subset of latent factors with sparse coefficients. We first generate a factor matrix $\boldsymbol{F} \in \mathbb{R}^{m \times k}$ whose columns denote the responses of $k$ linearly independent factors across $m$ unique stimuli. Each column is drawn i.i.d from $\mathcal{N}(0, 1)$, and the matrix is orthogonalized using the Gram-Schmidt procedure. We then generate a pair of representations

$Z_1 \in \mathbb{R}^{m \times n}$ and $Z_2 \in \mathbb{R}^{m \times n}$ as sparse linear mixtures of these factors:

$$X_1 = [X_1]_{ij} \sim \mathcal{N}(0,1), \quad X_2 = [X_2]_{ij} \sim \mathcal{N}(0,1)$$
$$M_1 = S \odot X_1, \quad M_2 = S \odot X_2 \quad \text{where} \quad S \in \{0,1\}^{k \times n}$$
$$Z_1 = FM_1, \quad Z_2 = FM_2$$

where $\odot$ denotes the Hadamard (element-wise) product. The binary mask $S$ controls the degree of sparsity. By using the *same* support mask for both populations, each column of $Z_1$ and the corresponding column of $Z_2$ depend on the same subset of factors but with independent mixing weights. Choosing $S$ to be highly sparse makes each "neuron" depend on only one (or a combination) of factors, thereby producing clear, approximately one-to-one correspondences across the two populations with the intent of mimicking selective tuning to a small set of features.

To introduce outliers and measurement noise, we augment each population with additional *"noise neurons"*. Concretely, we append $n'_1$ and $n'_2$ columns of i.i.d Gaussian noise $\varepsilon \sim \mathcal{N}(0,1)$, yielding $Z_1 \in \mathbb{R}^{m \times (n+n'_1)}$ and $Z_2 \in \mathbb{R}^{m \times (n+n'_2)}$.

### A1.4  BASELINE MATCHING ALGORITHMS

In this section, we describe the baseline matching algorithms used to rank-order the tuning similarities as demonstrated in Sec. 4.2. In all cases, we consider a representation pair $Z_1 \in \mathbb{R}^{m \times n_1}$ and $Z_2 \in \mathbb{R}^{m \times n_2}$ where $m$ is the number of stimuli and $\{n_1, n_2\}$ are the number of neurons respectively.

**Brute-Force Matching.** We construct a greedy baseline for ordering the deletion of neurons based on their soft-matching correlation score. Starting from a complete set of $N$ neurons, we establish a baseline score. At each iteration, we evaluate, for every remaining neuron $i$, the score obtained after removing neuron $i$ and re-fitting the soft-match transform on the reduced representation. We then remove the neuron whose deletion produces the largest decrease in the matching distance (equivalently, the largest improvement in the alignment score if removal improves the score), append it to the deletion order, and repeat on the remaining neurons. Thus, in essence, we construct a *rank-ordering* of neurons in terms of their tuning similarities based on the soft-matching objective.

---

**Algorithm 1** Brute-Force Matching

```
 1: R ← {1, …, N}                                              ▷ set of remaining neuron indices
 2: s̄ ← SoftMatch(Z_{1:,R}, Z_{2:,R})                               ▷ baseline score on full set
 3: π ← [ ]                                                      ▷ initialize deletion ordering
 4: while R ≠ ∅ do
 5:     for each i ∈ R do
 6:         R_{-i} ← R \ {i}
 7:         s_i ← SoftMatch(Z_{1:,R_{-i}}, Z_{2:,R_{-i}})        ▷ re-fit soft matching without neuron i
 8:         Δ_i ← s_i − s̄                                     ▷ change in score produced by deleting i
 9:     end for
10:     i* ← arg min_{i∈R} Δ_i                      ▷ pick neuron whose deletion most decreases score
11:     append i* to the end of list π
12:     R ← R \ {i*}                                             ▷ permanently remove neuron
13:     s̄ ← s_{i*}                                    ▷ update current score to the one after deletion
14: end while
15: return π                                        ▷ deletion order from least → most matched
```

---

**Correlation-Based Matching.** For each fitted soft-matching plan $T$ on a response pair, we perform a bidirectional correlation-based voxel matching. We project responses from one representational pair into the others space $\widetilde{Z_1} = Z_1 T$ and compute the Pearson correlation between the response pair $\mathrm{corr}(\widetilde{Z_1}, Z_2)$. We retain the top-$k$ correlated units in $Z_2$, where $k$ is determined by the number of units (un)matched using the partial soft-matching distance to maintain consistency during comparison. We repeat this procedure in the reverse direction $\widetilde{Z_2} = Z_2 T^\top$ and compute Pearson correlations $\mathrm{corr}(Z_1, \widetilde{Z_2})$ to find the matched units.

---

**Algorithm 2** Correlation-Based Matching

---

**Forward mapping (response 1 → response 2):**
1: $\widetilde{\boldsymbol{Z}}_1 \leftarrow \boldsymbol{Z}_1 \boldsymbol{T}$
2: $c_{1 \rightarrow 2} \leftarrow \texttt{corr}\left(\widetilde{\boldsymbol{Z}}_1, \boldsymbol{Z}_2\right)$
3: $\text{kept}_2 \leftarrow \texttt{argsort}(c_{1 \rightarrow 2})[:k]$
    **Reverse mapping (response 2 → response 1):**
4: $\widetilde{\boldsymbol{Z}}_2 \leftarrow \boldsymbol{Z}_2 \boldsymbol{T}^\top$
5: $c_{2 \rightarrow 1} \leftarrow \texttt{corr}\left(\boldsymbol{Z}_1, \widetilde{\boldsymbol{Z}}_2\right)$
6: $\text{kept}_1 \leftarrow \texttt{argsort}(c_{2 \rightarrow 1})[:k]$

---

**Partial Soft-Matching.** For a given regularization value $s$, we fit a partial soft-matching transport plan $\boldsymbol{T} \in \mathbb{R}^{n_1 \times n_2}$ between a response pair $\{\boldsymbol{Z}_1, \boldsymbol{Z}_2\}$. We compute the outgoing mass $r_i = \sum_i \boldsymbol{T}_{ij}$ for each source unit and incoming mass $c_j = \sum_j \boldsymbol{T}_{ij}$ for each target unit, and retain only those units whose total mass exceeds a small threshold $\tau$, serving as a way to determine the unmatched units. We repeat this procedure over a grid of regularization values $\mathcal{S} \leftarrow \{s_1, \cdots, s_k\}$, yielding "per-$s$" sets of *kept* units $\mathcal{K}_1(s)$ and $\mathcal{K}_2(s)$ that are matched.

---

**Algorithm 3** Partial Soft-Matching

---

1: $\mathcal{S} \leftarrow \{s_1, \cdots, s_k\}$          ▷ initialize a list of regularization values
2: $\tau \leftarrow \texttt{1e-6}$          ▷ initialize a threshold for transport weight
3: **for** $m \in \mathcal{M}$ **do**
4:      $\boldsymbol{T} \leftarrow \textsf{ParSM}(m_{\text{reg}} = s)$          ▷ compute optimal transport plan
5:      $r_i \leftarrow \sum_{j=1}^{n_2} \boldsymbol{T}_{ij}$ for $i = 1, \ldots, n_1$          ▷ outgoing mass per source unit
6:      $c_j \leftarrow \sum_{i=1}^{n_1} \boldsymbol{T}_{ij}$ for $j = 1, \ldots, n_2$          ▷ incoming mass per target unit
7:      $\mathcal{K}_1 \leftarrow \{ i \mid r_i \geq \tau \}$
8:      $\mathcal{K}_2 \leftarrow \{ j \mid c_j \geq \tau \}$
9:      **return** $\mathcal{K}_1, \mathcal{K}_2$
10: **end for**

---

## A1.5 SYNTHESIS OF MAXIMALLY EXCITING IMAGES

Given a CNN $f \colon \boldsymbol{\mathcal{X}} \to \mathbb{R}^K$ that maps an input image $\boldsymbol{x} \in \boldsymbol{\mathcal{X}} \subset \mathbb{R}^{h \times w \times c}$ to $K$ class logits, we define a scalar target $g(\boldsymbol{x})$ as the activation of the unit of interest (*e.g.,* feature-map channel, readout). For convolutional layers, when aligning representations, we evaluate channels at the *center* spatial location, motivated by evidence that convolutional feature maps are equivalent up to a circular shift (Williams et al., 2021; Kapoor et al., 2025). Fixing the center neuron across channels thus allows us to consistently describe representations. We synthesize one image per channel by solving:

$$\boldsymbol{x}^\star = \underset{\boldsymbol{x} \in \boldsymbol{\mathcal{X}}}{\arg\max} \left( \mathbb{E}_{\tau \sim \mathcal{T}} g(\tau(\boldsymbol{x})) - \sum_r \lambda_r R_r(\boldsymbol{x}) \right)$$

where $\mathcal{T}$ is a distribution over input transformations (*e.g.,* jitter, crop). Each $R_r(\cdot)$ is a regularizer with weight $\lambda_r$. In practice, we sample a new $\tau$ at each iteration and maximize the objective via gradient ascent at the center pixel of every channel. We implement this optimization with the Lucent library [2] using total variation (TV) as the regularizer. Each synthesized image is of shape $224 \times 224 \times 3$.

---

[2]https://github.com/TomFrederik/lucent/

## A1.6  PRIVILEGED AXES PERSISTS IN ALL KERNEL SUBPOPULATIONS

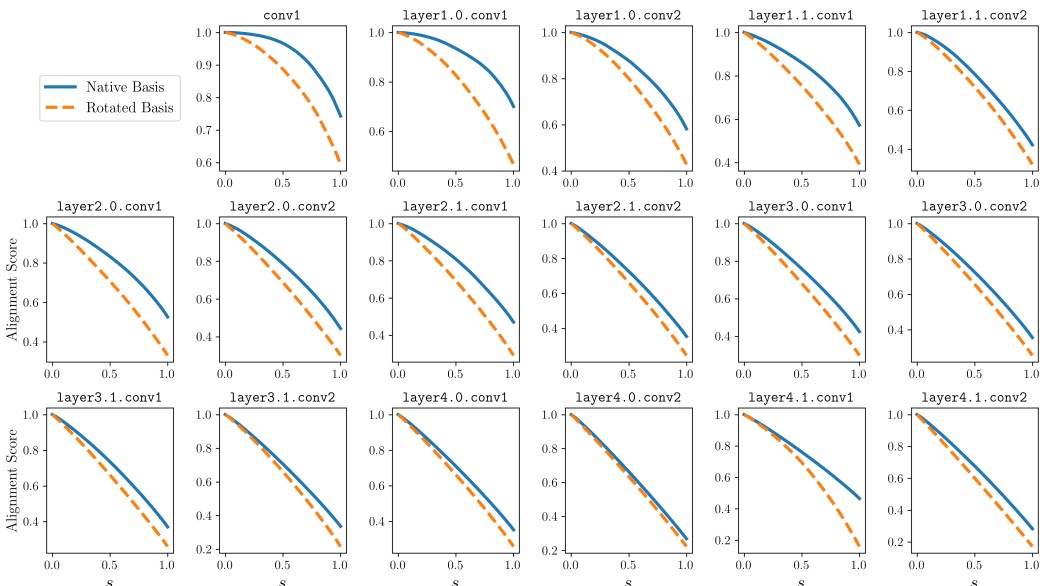

Figure A2: **Testing for Privileged Coordinate Systems Across Matched Neural Subpopulations.** For all convolutional layers in a pair of ImageNet-trained ResNet-18 models, we show that a privileged solution basis persists.

## A1.7  ADDITIONAL RESULTS FOR MATCHED MAXIMALLY EXCITING IMAGES

In the following section, we show additional matched MEI pairs for two layers in a ResNet-18, while still displaying the top $10\%$ and bottom $10\%$ examples in Fig. A3.

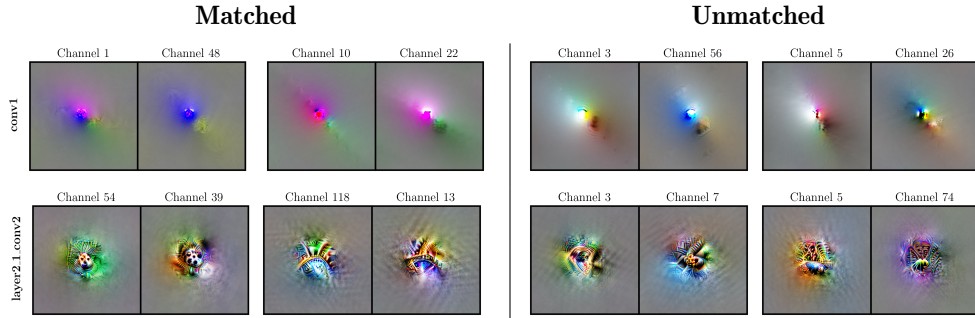

Figure A3: **Additional visualizations of (Un)matched Units Using Maximally Exciting Images.** We show additional MEIs for two layers of ResNet-18, comparing more pairs of top matched and unmatched units using the partial soft-matching metric.

## A1.8 Sensitivity To Choice of Cost Function

For all results demonstrated in Sec. 4.2, we use cosine similarity to rank-order units. In this section, we construct a squared-Euclidean distance cost matrix (i.e.: $C_{ij} = ||\boldsymbol{x}_i - \boldsymbol{y}_j||^2$) to compute the optimal transport plan. In synthetic experiments (Fig. A4-A), deep neural networks (Fig. A4-B) and brain data (Fig. A4-C), we find our conclusions remain unaffected by the choice of cost function. For the brain data and DNN alignment plots, we normalize the distances by their maximum value such that values can be visualized in the same plot.

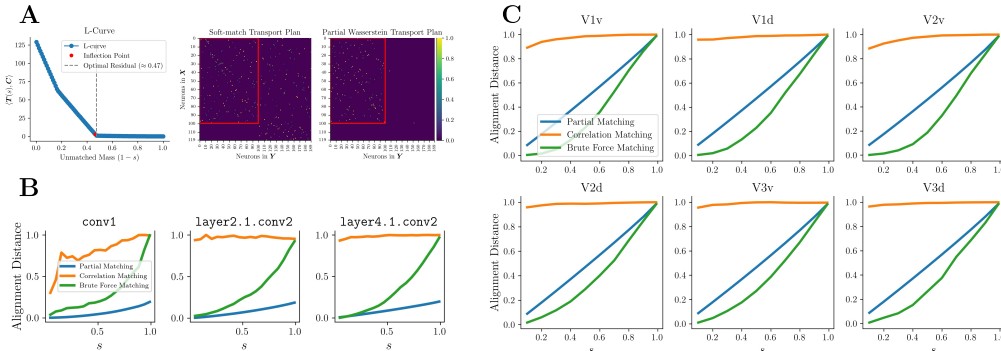

Figure A4: **Partial Soft-Matching Using a Euclidean Cost Function (A)** We visualize the L-curve elbow computed using a Euclidean cost function (**left**), and the corresponding transport plans (**right**). We find that the optimal regularization value $s$ and transport plans are identical to that computed using a cosine similarity cost function. **(B)** Rank-ordering of neurons from 3 layers (early, middle and late) in an ImageNet-trained ResNet-18 network using a squared Euclidean cost function. Removing units by using the Euclidean distance yields identical trends as using a cosine similarity cost. **(C)** We rank-order voxels from a subject pair in NSD using six visual areas by using a squared Euclidean cost function. Ordering voxels using a Euclidean distance cost function also reveals identical trends as using cosine similarity.