# OpenReview forum: "Partial Soft-Matching Distance For Neural Representational Comparison With Partial Unit Correspondence"
_ICLR.cc/2026/Conference — ICLR 2026 Poster_

### Official Review · Reviewer_QEdB · 2025-10-29

**Soundness:** 4
**Presentation:** 4
**Contribution:** 4
**Rating:** 8
**Confidence:** 4

**Summary:**

This paper proposes an extension of the soft matching distance for measuring representational alignment. The idea is to account for noisy, low-reliability units by relaxing the strict mass conservation of soft matching, thus allowing for unmatched units. The authors also provide a method for selecting the hyperparameter of the method, the optimal amount of unmatched mass. In simulation experiments, they show that their metric is able to select the better model of simulated data, while standard soft matching fails. They also test their method on real fMRI data (NSD), finding that it achieves higher ROI-matching accuracy than standard soft matching.

**Strengths:**

- The paper addresses a relevant question that is of great interest to the community.
- It adds a sensible extension to soft matching, creating a promising candidate for a good neural similarity measure. The proposed unbalanced soft matching is well-motivated and initial sanity checks are promising.
- The writing is clear and logical.
- I see no issues with quality or correctness.

**Weaknesses:**

- Arguably, the fact that there are unmatched units between two systems is a difference that we might not want to ignore. Suppose there were two essentially identical candidate models to be evaluated, but one has (many) random noise units added. Unbalanced soft matching with optimally chosen s (for each model) would not find a difference between these models, correct? Standard soft matching thus has an implicit bias for simpler models, which might be desirable, but is somewhat lost by your method (unless one uses the s to decide).
- On a similar note, the metric suffers from the fact that I can trivially achieve optimal similarity by just inflating my model: Because the metric is free to drop what it doesn't need, any sufficiently large model should eventually achieve perfect similarity (at potentially large values of s). This won't generalize to a test set, though.

Overall, while there is a potential weakness of the proposed method, I do think that the paper adds a valuable proposition to the research field of representational similarity.

**Questions:**

- To address the main criticism of the metric being too flexible: It might be necessary in practice to combine unbalanced soft matching with train-test split evaluations, where you choose s (and the units to be discarded) on the train split, to then compute the similarity score on the test split.
- Will you release a reference implementation to compute unbalanced soft matching?
- Section 4.3: How exactly does the matching of similar brain regions work? I currently understand that you select two ROIs in two subjects each, then compute a between-subject matching for all voxels of these regions, to then measure the fraction of units that were correctly mapped to the corresponding region in the other subject. Is this correct? The paper could be a bit more detailed here.
- For voxels of fMRI readings, there sometimes are uncertainty measures of individual voxels based on estimates of SNR. If you threshold units by this uncertainty measure, i.e. exclude them if they are too unreliable, does that match against the units your algorithm excludes? How do results in table change 1 if you compute soft matching but exclude unreliable units?
- How does $m_{reg}$ relate to $s$? Is it simply $m_{reg} = s$?
- I was initially confused about the notation $\mathcal{T}^{u}(N_x, N_y)$, because the $u$ is never mentioned. I now think this simply stands for "unbalanced", is that correct? Should this maybe be $\mathcal{T}^{s}(N_x, N_y)$? I would have found it logical, notation-wise, to include s in the reference to the set of natural solutions, because the set is parameterized in s.
- Figure 7 states 3 layers but I think it should say 2.
- Grammar in lines 192, 216.

---

> ### Author Response · Authors · 2025-11-21
>
> We would like to sincerely thank reviewer **QEdB** for their positive feedback on our manuscript, and noting the relevance of our proposed method. We address each of their concerns below:
>
> > Arguably, the fact that there are unmatched units between two systems is a difference that we might not want to ignore...
> The reviewer touches on an important philosophical question about what our representational similarity measures should be invariant to: should they be invariant to extra noise dimensions or spurious units? We argue that the answer depends critically on the application context and the nature of the data being compared.
>
> We agree that the number of matched versus unmatched units contains meaningful information about model similarity. However, we argue this information is not lost but rather made explicit and interpretable through our framework. Critically, standard soft-matching obscures the distinction between core computational differences and noise differences by forcing all units into correspondence. In the reviewer's example of comparing two identical models where one contains additional random noise units, standard soft-matching would report degraded alignment, but this could reflect either: **(1)** genuine computational differences in the core units, or **(2)** the presence of noise units that must be spuriously matched. Our method disentangles these scenarios: the alignment score reflects the quality of true correspondences, while s indicates the fraction of the population participating in the correspondence.
>
> This distinction becomes crucial in practical neural recording settings, where measurements are frequently corrupted by noise from device artifacts (*e.g.,* faulty electrodes, motion artifacts), inactive recording channels, or voxels/electrodes that inadvertently sample from neighboring anatomical areas due to imprecise boundary definitions. When such outlier or non-corresponding responses must be matched as is unavoidable under standard soft-matching they risk obscuring the underlying representational similarity between neural populations that are otherwise well-aligned in their functional responses.
>
> A concrete example illustrates this point. Consider two putatively corresponding brain regions whose anatomical boundaries are imperfectly defined due to individual variability or registration errors. Standard soft-matching would enforce correspondence between all voxels, including those at anatomical boundaries that may belong to distinct functional regions. This forced matching of non-corresponding units would suppress the alignment score. In such scenarios, the resultant drop in similarity does not necessarily reflect a true computational difference, but rather a consequence of suboptimal voxel inclusion. Partial matching, by allowing units to remain unmatched, isolates the homologous subpopulations thereby yielding a more faithful measure of representational similarity for the truly corresponding neural tissue.
>
> > On a similar note, the metric suffers from the fact that I can trivially achieve optimal similarity by just inflating my model: Because the metric is free to drop what it doesn't need, any sufficiently large model should eventually achieve perfect similarity (at potentially large values of s). This won't generalize to a test set, though.
>
> The reviewer raises a legitimate concern about selecting $s$ through cross-validation. Indeed, such a procedure would be problematic: maximizing alignment on held-out data would almost certainly yield extremely small $s$ values (dropping most mass) to achieve near-perfect alignment among the few remaining highly-correlated units. Importantly, this pathology is not specific to inflated models; it could occur for any pair of neural representations when $s$ becomes sufficiently small. This is precisely why we do not advocate selecting $s$ via cross-validation, but instead propose alternatives such as the L-curve heuristic or computing the area under the alignment-regularization curve across multiple $s$ values.
>
> However, we emphasize that a primary use case of our method—demonstrated extensively in **Sections 4.2**, **4.4**, and **4.5** is obtaining computationally efficient rankings of units by their alignment strength (identifying most versus least matched tuning functions). This application does not require selecting a single optimal s value, but rather leverages the method's ability to partition populations by correspondence quality. For these analyses, which constitute the dominant practical application in neuroscience and machine learning contexts, the reviewer's concern about achieving "trivial" perfect similarity through selection of small $s$ does not apply.

---

> ### Author Response · Authors · 2025-11-21
>
> > Will you release a reference implementation to compute unbalanced soft matching?
>
> Yes, we plan to release our code publicly at the end of the review period.
>
> > Section 4.3: How exactly does the matching of similar brain regions work? I currently understand that you select two ROIs in two subjects each, then compute a between-subject matching for all voxels of these regions, to then measure the fraction of units that were correctly mapped to the corresponding region in the other subject. Is this correct? The paper could be a bit more detailed here.
>
> Yes, the reviewer’s understanding is correct. We agree that our original description of the experimental procedure in Sec. 4.3 is overly terse and may have been difficult to follow. We have now revised this section to be slightly more verbose (**L401**-**L403**). We hope that these changes make our experimental setup clear.
>
> > For voxels of fMRI readings, there sometimes are uncertainty measures of individual voxels based on estimates of SNR. If you threshold units by this uncertainty measure, i.e. exclude them if they are too unreliable, does that match against the units your algorithm excludes? How do results in table change 1 if you compute soft matching but exclude unreliable units?
>
> The reviewer raises a good point. We have now repeated the same experiments by thresholding based on the noise ceilings for the voxels over two different values $\lbrace 0.1, 0.3\rbrace$. We tabulate these results in **Table 1** and observe that there is an expected, high-degree of correlation between the fraction of units kept and their corresponding precision: stricter noise-ceiling thresholds increase precision but at the cost of discarding a substantial fraction of voxels. Despite this, we find that unbalanced soft-matching, for most cases, yields a higher precision while keeping most voxels to compute correspondence. This indicates that UnSM is not simply replicating a noise-ceiling filter; the units excluded by UnSM may largely be those that noise-ceiling thresholds would remove, but UnSM achieves this adaptively and with better precision–retention trade-offs.
>
> > How does m_reg relate to s? Is it simply s?
>
> Indeed. We apologize for the confusion caused by changing our notation in different parts of the manuscript. We have now corrected this to say $s$ everywhere and ensure consistency throughout—both in the text and all figures.
>
> > I was initially confused about the notation T^u(N_x, N_y), because the u is never mentioned. I now think this simply stands for "unbalanced", is that correct? Should this maybe be T^s(N_x, N_y)? I would have found it logical, notation-wise, to include s in the reference to the set of natural solutions, because the set is parameterized in s.
>
> We appreciate the reviewer's suggestion; this is indeed a cleaner and more succinct notation. We have updated the manuscript to denote the transportation polytope in the partial OT setting as $\mathcal{T}^s(N_x, N_y)$ throughout.
>
> > Figure 7 states 3 layers but I think it should say 2.
> Grammar in lines 192, 216.
>
> Thank you for catching these! We have now made edits to correct this.

---

### Official Review · Reviewer_SBuW · 2025-11-01

**Soundness:** 3
**Presentation:** 2
**Contribution:** 4
**Rating:** 6
**Confidence:** 2

**Summary:**

The paper extends soft matching for neuron to neuron alignment to a partial optimal transport setting that can leave units unmatched. It is rotation sensitive (in some cases, a desired quality) yet robust to noise and unequal population sizes. They use simulations, fMRI across NSD subject pairs, and DNN experiments to show that the method filters out low-quality units, improves cross-area precision, and closely tracks brute force ablations at far lower cost.

**Strengths:**

In summary, I think the originality and significance of this paper are good.

- I believe the paper addresses an important and timely problem.

- Recasting as average matched correlation gives an interpretable score in [−1,1], which helps.

- Experiments seem thorough. Simulations show recovery with outliers and correct model selection. NSD shows exclusion of low noise ceiling voxels and higher within-area precision than balanced soft matching. DNNs show matched units have similar MEIs and that alignment drops after random rotations, consistent with privileged axes.

**Weaknesses:**

- Novelty relative to existing unbalanced OT and partial matching methods could be clearer. The authors cite Chapel et al. and related work, but I am not sure what is gained over the existing partial OT with dummy nodes plus a simple threshold.

- My understanding is that the tuning curves are centered as unit-normalized before being used for the unbalanced soft-matching. However, there are many cases where the neuronal units have very small responses (dead neurons). This normalization scheme would blow up the activities of these dead neurons, and I think that would affect the matching result. I think the authors should address the sensitivity of UnSM to the preprocessing methods.

- The clarity of the paper can be improved. Readers would appreciate a more detailed explanation of the original soft-matching algorithm at the beginning. The authors can use that opportunity to define variables more carefully. Currently, variables such as p, and q, and terminologies like "mass" are not defined clearly before being used. The mathematical part of the paper is currently only really readable to readers who are already familiar with the OT literature or the softmatching paper.

**Questions:**

1. The variables p and q are used in line 108 without being explained or introduced.

2. For choosing optimal regularization, the authors first introduce a 2D-valued function f(s)=[\zeta(s), \rho(s)], with a justification that looking at both quantities is necessary for balancing the transport distance and reg. strength. But then the authors proceed to ignore \rho(s) when computing the maximal positive curvature. Why is that?

3. I don’t think I understand the Figure 2 left image. What is the red box and the yellow line? If both X and Y have 100 signal neurons, then shouldn’t the matching combinations form a square submatrix of size 100x100?

---

> ### Author Response · Authors · 2025-11-21
>
> We would like to thank reviewer **SBuW** for identifying the importance of the problem our proposed metric addresses. Below, we address each of their concerns:
>
> > Novelty relative to existing unbalanced OT and partial matching methods could be clearer. The authors cite Chapel et al. and related work, but I am not sure what is gained over the existing partial OT with dummy nodes plus a simple threshold.
>
> We acknowledge that partial optimal transport and exact solutions to the partial $p$-Wasserstein problem have been studied rigorously and developed under other applications (*e.g.,* positive-unlabeled learning). However, our contribution lies not in inventing partial OT, but in recognizing and demonstrating its utility for neural representational comparison where the objects being compared are neural tuning functions. Specifically, our work addresses scenarios ubiquitous in neuroscience and deep learning where partial correspondence between units is expected: noisy or inactive neurons in biological recordings, model-specific computational units in DNNs, and outlier/task-irrelevant units that can disproportionately inflate distances. The connection between soft-matching distance and Wasserstein distance where soft-matching is the $2$-Wasserstein distance between uniform mixtures of Dirac masses at neural response vectors enables rotation-sensitive unit-level comparisons between neural representations. No prior work has applied partial OT to the representational-alignment problems we study, nor explored how partial correspondence enables robustness to noise, partitioning populations by correspondence quality, enabling both efficient rank-ordering of units by alignment strength and focused analyses on well-matched subpopulations (*e.g.,* testing for privileged axes only among aligned units). Moreover, we demonstrate that this metric is a powerful tool for cross-brain comparisons and is particularly valuable when anatomical boundaries are ill-defined, as is often the case with biological datasets.
>
> > My understanding is that the tuning curves are centered as unit-normalized before being used for the unbalanced soft-matching. However, there are many cases where the neuronal units have very small responses (dead neurons). This normalization scheme would blow up the activities of these dead neurons, and I think that would affect the matching result. I think the authors should address the sensitivity of UnSM to the preprocessing methods.
>
> We agree that this is a legitimate concern in scenarios where one uses an unbounded distance (*e.g.,* Euclidean distance). Very large/small activation values can inflate/deflate individual entries of the cost matrix $\boldsymbol{C}$, which in turn inflates the transport cost $\sum{\boldsymbol{TC}}$, and can make a few outlier activations dominate the alignment. However, since we optimize over the set of admissible couplings using a Pearson correlation, all entries of the cost matrix are necessarily bounded in $[-1, 1]$. Thus, the entries of $\boldsymbol{C}$ are bounded regardless of the raw activation magnitudes. Hence, dead units would not affect the matching quality if one makes use of a bounded cost function during the optimization routine.
>
> > The clarity of the paper can be improved. Readers would appreciate a more detailed explanation of the original soft-matching algorithm at the beginning. The authors can use that opportunity to define variables more carefully. Currently, variables such as p, and q, and terminologies like "mass" are not defined clearly before being used. The mathematical part of the paper is currently only really readable to readers who are already familiar with the OT literature or the softmatching paper.
>
> We thank the reviewer for their valuable suggestions. In the revised manuscript, we now explicitly define $\boldsymbol{p}$ and $\boldsymbol{q}$ as empirical probability measures supported on sets of uniform Dirac point masses $\mathcal{X}={\lbrace\boldsymbol{x}_i\rbrace}$ and $\mathcal{Y}={\lbrace\boldsymbol{y}_j\rbrace}$. We also clarify our use of “mass”: the total mass of a measure is its $\ell_1$-norm, $||\boldsymbol{p}||_1$ and $||\boldsymbol{q}||_1$. In standard OT formulations, the linear equality constraints enforce $||\boldsymbol{p}||_1 = ||\boldsymbol{q}||_1$. By contrast, the partial OT problem relaxes the equality constraint, and transports only a fraction $s$ of the total mass. Concretely, we solve for the cheapest transport plan that moves a mass $0\leq s\leq \min(||\boldsymbol{p}||_1,  ||\boldsymbol{q}||_1)$ between $\boldsymbol{p}$ and $\boldsymbol{q}$. This formulation and definition for “mass” are now explicitly stated in **L107**-**L111**. We have also made the description of soft-matching (**Sec. 2.1**) more verbose to address readers who may not be familiar with the method.

---

> > ### Author Response · Authors · 2025-11-21
> >
> > > The variables p and q are used in line 108 without being explained or introduced.
> >
> > We have addressed this concern in our response to the previous question.
> >
> > > For choosing optimal regularization, the authors first introduce a 2D-valued function f(s)=[\zeta(s), \rho(s)], with a justification that looking at both quantities is necessary for balancing the transport distance and reg. strength. But then the authors proceed to ignore \rho(s) when computing the maximal positive curvature. Why is that?
> >
> > We apologize for the confusion caused by our notation. The current presentation incorrectly suggests that $\rho(s)$ is not used while computing maximal curvature. In practice, however, the second centered finite difference $\delta^2$ is always evaluated with respect to the regularization strength parametrized by $\rho(s)$. To remove this ambiguity, we have now revised the notation in the manuscript (**L168**-**L170**), replacing $\delta^2$ with $\delta^2_\rho$ to make this dependence explicit. We hope that these changes alleviate the reviewer’s concern.
> >
> > > I don’t think I understand the Figure 2 left image. What is the red box and the yellow line? If both X and Y have 100 signal neurons, then shouldn’t the matching combinations form a square submatrix of size 100x100?
> >
> > The red box denotes neurons in representation $\boldsymbol{X}$ that are unmatched in $\boldsymbol{Y}$. The yellow feature in Fig. 2 (**left**) was inadvertently drawn as a line—it should be a yellow box aligned with the transport-plan axis, indicating neurons in $\boldsymbol{Y}$ that are unmatched in $\boldsymbol{X}$.
> >
> > For clarity—the unmatched block is indeed a square submatrix of dimension $100\times 100$. The apparent asymmetry in the original figure arises because the full matrix has different axis lengths ($120$ rows, $190$ columns), which visually elongates one dimension relative to the other. Hence, we understand the reviewer's concern, since the figure may give the reader a misleading impression that one of the matched regions was longer than the other.
> >
> > We have now updated the figure in the revised manuscript to show **only** the signal neurons (red box), and added tick labels to each of the transport plan axes for disambiguating any concerns with the shape of the transport plan.

---

### Official Review · Reviewer_Ny7g · 2025-11-07

**Soundness:** 3
**Presentation:** 2
**Contribution:** 3
**Rating:** 6
**Confidence:** 4

**Summary:**

Existing representational similarity metrics frequently fall into two categories: rotation-invariant methods (e.g., CKA, RSA) that measure overall geometric similarity but obscure neuron-level correspondence, or rotation-sensitive methods (e.g., standard soft-matching) that find such correspondences but are brittle, as they inherit the "critical limitation" of classical optimal transport (OT) by forcing a complete, one-to-one match between all units. This paper identifies the latter's inadequacy in the common scenario of comparing noisy biological data (fMRI) or functionally distinct deep neural networks (DNNs), where correspondence is inherently partial. It introduces the "unbalanced soft-matching distance," a principled extension of soft-matching to a partial OT framework. This method permits a fraction of units to remain unmatched, governed by a data-driven regularization parameter selected via an L-curve heuristic. The authors demonstrate through simulations that the method is robust to spurious "outlier" neurons and correctly identifies models with greater signal overlap in a task where standard soft-matching fails.

**Strengths:**

1. The method is theoretically well-grounded, providing a principled extension of optimal transport to solve the well-documented problem of spurious alignments in noisy, partially corresponding neural data.
2. The paper provides compelling evidence of practical utility by showing that a single $\mathcal{O}(n^3 \log n)$ optimization can rank unit alignment as accurately as a computationally prohibitive $\mathcal{O}(n^4 \log n)$ brute-force ablation.
3. The method provides highly interpretable outputs by partitioning units into "matched" and "unmatched" subsets.

**Weaknesses:**

1. The method's autonomy relies entirely on an L-curve heuristic to select the optimal matched mass, a technique the authors concede has unclear generality and is known to be unstable in non-ideal (e.g., smooth) cost-regularization landscapes.
2. By relaxing mass conservation, the proposed method sacrifices formal metric properties, specifically the triangle inequality, limiting its use in downstream algorithms or theoretical proofs that require a true metric.
3. The "correlation-based ordering" heuristic used as a fast baseline appears to be a strawman, as its catastrophic failure is predictable; a more competitive baseline, such as a greedy matching algorithm, was not included for comparison.

**Questions:**

1. The method relies on a squared Euclidean distance cost (Section 2.2). How sensitive are the resulting unit partitions and alignment rankings to this choice, and have alternative cost functions (e.g., cosine distance) been investigated?
2. How does the L-curve heuristic's second-derivative approximation perform on smooth cost-regularization curves that lack a distinct "elbow," and what is the method's failure mode in such cases?

---

> ### Author Response · Authors · 2025-11-21
>
> We would like to thank reviewer **Ny7g** for their positive assessment. We address each of the reviewers questions and concerns below:
>
> > The method's autonomy relies entirely on an L-curve heuristic to select the optimal matched mass, a technique the authors concede has unclear generality and is known to be unstable in non-ideal (e.g., smooth) cost-regularization landscapes.
>
> We appreciate the reviewer's attention to the L-curve heuristic. While we acknowledge its limitations, we note three important points:
>
> First, the L-curve is widely used across scientific domains (regularization theory, inverse problems). While this heuristic has known limitations, it provides a principled approach for balancing the tradeoff we face in partial transport: alignment quality (lower transport cost among matched units) versus coverage (fraction of units matched). The elbow identifies where we experience diminishing returns from excluding additional units i.e., where forcing more matches begins to substantially degrade the average quality of those matches. Our empirical results demonstrate that this heuristic performs robustly across diverse settings (simulations, fMRI data) where the optimal regularization varies substantially.
>
> Second, a primary use case of our method is obtaining a computationally efficient mechanism for rank-ordering units by their alignment strength (most vs. least matched tuning functions). This application which we demonstrate extensively in **Sections 4.2**, **4.4**, and **4.5** does not require optimizing over a singular choice of regularization parameter $s$.
>
> Third, we have now added a best practices section (**Appendix A1.1**) in response to reviewer feedback, explicitly noting that users should visually inspect the cost-regularization curve and override automatic selection when appropriate, particularly when the curve is smooth and without a clear elbow. This flexibility mirrors standard practice with hyperparameter selection in ML pipelines.  As we note in the manuscript (**L521**), alternative rank-based summaries such as area under the curve over matched mass (alignment-regularization curve) could potentially be a useful alternative to the L-curve heuristic as it aggregates information across all regularization values rather than relying on a single "optimal" choice
>
> > By relaxing mass conservation, the proposed method sacrifices formal metric properties, specifically the triangle inequality, limiting its use in downstream algorithms or theoretical proofs that require a true metric.
>
> We appreciate the reviewer highlighting this theoretical limitation. We already note this explicitly in the manuscript. However, we emphasize that:
> First, satisfying the triangle inequality is primarily useful for analyses involving large-scale network clustering or multi-way comparisons. For the dominant use cases in comparative analyses such as comparing one reference system (*e.g.,* brain recordings) against multiple candidate models, or rank-ordering units by alignment strength between two representations - triangle inequality violations have minimal practical impact.
>
> Second, sacrificing metric axioms is not unprecedented in representational similarity analysis. Widely used measures like linear predictivity do not satisfy metric axioms such as symmetry yet multiple studies successfully employ symmetrized versions despite lacking full “metricity”. The practical utility of a measure often outweighs strict adherence to metric axioms.
>
> Third, we note that recent work **[1]** has developed partial Wasserstein variants that do satisfy metric properties, including the triangle inequality. Future extensions of our framework could incorporate these formulations for applications requiring strict metric properties, though they are unnecessary for our primary use cases of unit ranking and pairwise alignment quality assessment. We now discuss this in lines **L524**-**L527**:
>
> *“Because partial OT relaxes mass conservation and violates the triangle inequality, it should be understood as a comparative tool rather than a strict metric. However, we note that recent theoretical work has developed partial Wasserstein variants that preserve full metric properties, including the triangle inequality (Raghvendra et al., 2024). Future extensions could integrate these formulations for applications requiring strict metric axioms, such as clustering analyses”*

---

> ### Author Response · Authors · 2025-11-21
>
> > The "correlation-based ordering" heuristic used as a fast baseline appears to be a strawman, as its catastrophic failure is predictable; a more competitive baseline, such as a greedy matching algorithm, was not included for comparison.
>
> We welcome the reviewer’s concern and would appreciate clarification on why they expect correlation-based ordering to perform catastrophically in this setting.
> For completeness, we restate what our correlation-based heuristic actually does, since its purpose may have been unclear. After fitting a single global soft-matching plan $\boldsymbol{T}$ on the full population, we project each population into the other’s representational space $(\widetilde{\boldsymbol{Z}}_1 = \boldsymbol{Z}_1\boldsymbol{T}$ and $\widetilde{\boldsymbol{Z}}_2 = \boldsymbol{Z}_2\boldsymbol{T}^\top)$. We then compute the Pearson correlation between $(\widetilde{\boldsymbol{Z}}_1, \boldsymbol{Z}_2)$ and symmetrically, $({\boldsymbol{Z}}_1, \widetilde{\boldsymbol{Z}}_2)$ and rank units by these correlations. Units with the smallest correlations are treated as “unmatched” or low-importance.
> We don’t think this baseline is a strawman; it is a natural approach practitioners might attempt. The intuition is straightforward: if a neuron has low correlation with its assigned partner in the soft-matching transport plan, this should indicate poor alignment with the population overall, making it a candidate for exclusion. Its empirical failure suggests that the global optimal transport structure cannot be reduced to local pairwise correlation statistics. A neuron might exhibit high correlation with its assigned partner yet be dispensable for overall alignment because other neurons compensate for its removal. Conversely, a neuron with moderate pairwise correlation might be critical to the global matching structure. This failure mode is not obvious or expected a priori, which is precisely why we include this baseline.
>
> Regarding the reviewers concern about a “greedy baseline”: our manuscript already implements a “brute-force baseline”, which we allude to in our manuscript as “Brute-Force Matching” (**Figs. 5**, **6** and pseudocode in **Appendix A.1.3**) as it relies on exhaustive re-optimization to provide the ground-truth ranking. To reiterate, in this approach, we remove each candidate neuron in turn and measure the resulting drop in alignment score. The neuron whose deletion elicits the largest drop is taken to be the most aligned unit. We iteratively perform this operation over the entire set of neurons to construct a rank-ordering.
>
> > The method relies on a squared Euclidean distance cost (Section 2.2). How sensitive are the resulting unit partitions and alignment rankings to this choice, and have alternative cost functions (e.g., cosine distance) been investigated?
>
> We use a cosine-based cost, specifically, the **cosine similarity**. Equivalently, this is the Pearson correlation computed on mean-centered, unit-normalized responses as the pairwise cost function used for alignment (Sec. 2.4). That said, we note that our current wording in Sec. 2.2 can be easily misinterpreted to imply that we make use of a squared Euclidean cost. To address this, we have now revised the description in **Sec. 2.2** to make the cosine formulation explicit. We now conduct our experiments to rank-order units using a squared-Euclidean cost function and found no qualitative or quantitative change to our conclusions. We include these additional results in the supplementary (**Appendix. A1.7**).
>
> > How does the L-curve heuristic's second-derivative approximation perform on smooth cost-regularization curves that lack a distinct "elbow," and what is the method's failure mode in such cases?
>
> As noted in our earlier response, we frame our partial matching method as a means to perform relative comparisons (*e.g.,* rank-orderings) of units across neural representations, not as a strict metric. For most of the experiments (**Sec. 4.1**, **4.2**, **4.4**, **4.5**), one does not require the L-curve heuristic to select a mass regularization parameter, and thus obtain said rank-orderings. Nevertheless, we agree with the reviewer that the L-curve heuristic fails in smooth landscapes without a clear elbow, and these cases should be stated explicitly. We have now added a “Best Practices” section in our supplementary **Appendix. A.1.1** to:
> - Caveat the L-curve as a user-dependent choice, akin to the choice of hyperparameters common in machine learning applications.
> - Describe the observed failure modes—specifically, when the cost-regularization curve is smooth and monotonic. In such a scenario, the inflection point determined by finding the 2nd finite difference is merely an artifact of the smoothness of the curve. In these cases, one typically observes that the “optimal” regularization appears to be at the tail ends of the $(\zeta(s), \rho(s))$ curve, which would warrant additional visual checks.

---

> ### Author Response · Authors · 2025-11-21
>
> References:
>
> **[1]** A New Robust Partial $p$-Wasserstein-Based Metric for Comparing Distributions (Raghvendra et al., 2024)

---

> > ### Comment · Reviewer_Ny7g · 2025-11-27
> >
> > Thank the authors for their efforts on rebuttals. Most of my concerns are addressed. I tend to keep my score.

---

### Author Response · Authors · 2025-12-04
**Summary of modifications, thanks to all reviewers for constructive feedback**

We thank all the reviewers for their valuable comments and suggestions to help improve our manuscript. We summarize key changes since our initial submission below:

1. In response to reviewer **Ny7g**, we note that the proposed L-curve heuristic used for selecting the mass regularization parameter $s$, can in certain situations (*e.g.,* smooth monotonic cost-regularization curves) be an ambiguous choice. To address this, we have now added a **Best Practices** section in **Appendix A1.1**, where we explicitly note the failure modes of this method, suggest simple diagnostics to the practitioner such as visual verification of the picked regularization value and discuss alternatives such as the area under the alignment-regularization curve. Moreover, we also note that the core application of our method for **rank-ordering units** based on alignment strength remains **unaffected** by the L-curve heuristic. These additions are intended to present the L-curve as a useful heuristic, analogous to the choice of hyperparameters in any typical machine learning application.

2. Reviewer **Ny7g** also raised concerns about the sensitivity of the method to alternative cost functions. To address this, we re-ran a subset of experiments using a **squared-Euclidean cost function** (**Appendix A1.8**) in addition to cosine/Pearson cost which we have used throughout our manuscript. We note that both our qualitative (*e.g.,* unit rank-ordering) and quantitative conclusions remain unchanged.

3. Reviewer **SbuW** pointed out concerns regarding the introduction of notation without explicitly defining what constitutes mass and empirical distributions, making the manuscript harder to follow for readers not familiar with soft-matching and optimal transport theory. We have updated the introduction paragraph in **Sec. 2** and **Sec 2.1** to address these concerns, and hope that the manuscript is now readable to a wider audience.

4. In response to reviewer **SbuW**, we have also updated **Fig. 2** of our manuscript for clarity—we agree that in its original framing, it is indeed much harder to comprehend. We have now updated the figure in the revised manuscript to show only the signal neurons, and include tick labels along the transport plan axes to avoid any visual misinterpretations of block shapes and aspect ratios.

5. Next, reviewer **QEdB** raises some important (philosophical) questions—should representational metrics be invariant to extra noise dimensions or spurious units? We argue that the answer depends critically on the application context and the nature of the data being compared. To illustrate, we discuss a concrete example: when anatomical boundaries are imperfect, forcing one-to-one correspondences across all voxels can spuriously suppress alignment by matching non-homologous voxels. Partial matching (with $s<1$) explicitly allows unmatched units and thus isolates the genuinely corresponding subpopulations, providing a more faithful measure of similarity for the homologous neural tissue.

6. In response to reviewer **QEdB**, we also add a new set of results in **Table 1**, where we compute soft-matching on a subset of voxels which are thresholded based on the degree of signal noise recorded. We find that unbalanced soft-matching (UnSM), for most cases yields a higher precision whilst keeping most voxels to compute correspondence—inidicating that UnSM isn’t merely a noise-ceiling filter but a much better tool for adaptively matching voxels with genuine correspondence.

7. Last, we would like to thank reviewer **QEdB** for painstakingly proofreading our manuscript for grammatical errors and typos, which we greatly appreciate and have corrected in our revised manuscript.

We now also add a new set of supplementary MEI matching examples (**Appendix A.1.7**) so readers can verify that the reported cases in Sec. 4.4 are representative and not cherry-picked. We hope that these changes make our manuscript better and readable to a wider audience, as well as address any scientific concerns.

---

### Meta-Review · Area_Chair_JCq4 · 2025-12-15

**Summary:**

This paper extends soft-matching distance to a partial optimal transport framework, allowing neural units to remain unmatched.

Reviewers raised three main categories of concerns:

1. Methodological: The L-curve heuristic for selecting the regularization parameter was questioned for generality and stability. The relaxation of mass conservation sacrifices the triangle inequality, limiting use in clustering or multi-way comparisons.
2. Conceptual: A philosophical question was raised about whether representational metrics should penalize extra noise dimensions—unbalanced soft-matching could report high similarity between models where one has many spurious units added.
3. Presentation: Initial notation was unclear, with variables like p, q, and "mass" used before definition, making the method inaccessible to readers unfamiliar with optimal transport. Figure 2 was confusing regarding the structure of matched/unmatched blocks.
* Minor concerns included sensitivity to cost function choice, preprocessing effects on dead neurons, and novelty relative to existing partial OT methods.

**Reviewer Concerns:**

Addressed:

* L-curve limitations: Authors added a "Best Practices" appendix explicitly documenting failure modes and diagnostic procedures, and clarified that the primary use case (unit ranking) doesn't require optimizing a single parameter value.
* Cost function sensitivity: New experiments with squared-Euclidean cost (Appendix A1.8) showed qualitatively unchanged results.
* Presentation clarity: Section 2 was revised to define p, q, and "mass" explicitly; Figure 2 was redesigned with only signal neurons shown and axis labels added.
* Noise ceiling analysis: New Table 1 results show UnSM achieves better precision-retention tradeoffs than simple thresholding.
* Philosophical concern: Authors provided an explanation that forced matching in noisy data can obscure true correspondence, and that the number of matched units remains interpretable.

Outstanding/Acknowledged Limitations:

* Triangle inequality violation: The authors acknowledged this as a limitation, noting that the method should be understood as a comparative tool rather than a strict metric.
* Model inflation concern (QEdB): While the authors explained why cross-validation for s selection would be problematic, the fundamental issue that sufficiently large models can achieve perfect similarity remains.

**Reviewer Scores:**

* Reviewer Ny7g (original score 6): Would likely maintain score at 6. They explicitly stated after rebuttal: "Most of my concerns are addressed. I tend to keep my score."

* Reviewer SBuW (original score 6): May have increased to 7. They raised primarily presentation and clarity concerns, all of which were addressed: notation was clarified, Figure 2 was redesigned, and the dead neuron concern was resolved by explaining the bounded nature of correlation-based costs.

* Reviewer QEdB (original score 8): Would likely maintain score at 8. Apart from the philosophical question, the authors addressed their question by new noise ceiling experiments (Table 1).

---

### Decision · Program_Chairs · 2026-01-26

Accept (Poster)